# Autosomal Dominant Polycystic Kidney Disease: Is There a Role for Autophagy?

**DOI:** 10.3390/ijms241914666

**Published:** 2023-09-28

**Authors:** Claudio Ponticelli, Gabriella Moroni, Francesco Reggiani

**Affiliations:** 1Independent Researcher, via Ampère 126, 20131 Milan, Italy; ponticelli.claudio@gmail.com; 2Nephrology and Dialysis Unit, IRCCS Humanitas Research Hospital, Via Manzoni 56, 20089 Milan, Italy; gabriella.moroni@hunimed.eu; 3Department of Biomedical Sciences, Humanitas University, 20090 Milan, Italy

**Keywords:** autosomal dominant polycystic kidney disease, autophagy, renal involvement, polycystins, kidney diseases

## Abstract

Autosomal-Dominant Polycystic Kidney Disease (ADPKD) is a monogenic disorder initiated by mutations in either *PKD1* or *PKD2* genes, responsible for encoding polycystin 1 and polycystin 2, respectively. These proteins are primarily located within the primary cilia. The disease follows an inexorable progression, leading most patients to severe renal failure around the age of 50, and extra-renal complications are frequent. A cure for ADPKD remains elusive, but some measures can be employed to manage symptoms and slow cyst growth. Tolvaptan, a vasopressin V2 receptor antagonist, is the only drug that has been proven to attenuate ADPKD progression. Recently, autophagy, a cellular recycling system that facilitates the breakdown and reuse of aged or damaged cellular components, has emerged as a potential contributor to the pathogenesis of ADPKD. However, the precise role of autophagy in ADPKD remains a subject of investigation, displaying a potentially twofold impact. On the one hand, impaired autophagy may promote cyst formation by inducing apoptosis, while on the other hand, excessive autophagy may lead to fibrosis through epithelial to mesenchymal transition. Promising results of autophagy inducers have been observed in preclinical studies. Clinical trials are warranted to thoroughly assess the long-term safety and efficacy of a combination of autophagy inducers with metabolic and/or aquaferetic drugs. This research aims to shed light on the complex involvement of autophagy in ADPKD, explore the regulation of autophagy in disease progression, and highlight the potential of combination therapies as a promising avenue for future investigations.

## 1. Autosomal-Dominant Polycystic Kidney Disease

Autosomal-Dominant Polycystic Kidney Disease (ADPKD) stands as the most prevalent monogenic disorder impacting individuals across all global ethnicities, with an incidence that varies around 1 in 1000 [1]. The disease is characterized by the formation of multiple cysts gradually compressing renal parenchyma and eventually leading to end-stage kidney disease (ESKD) in adulthood. ADPKD is caused by mutations in *PKD1* or *PKD2* genes. The two disease loci are on separate chromosomes and segregate independently. The *PKD1* gene is located on chromosome 16p13.3 and encodes polycystin 1 (PC1). *PKD2* gene is located on chromosome 4q22.1 and encodes polycystin 2 (PC2) [2,3,4]. The disease is characterized by large interfamilial and intrafamilial variability due to genetic heterogeneity. About 80–85% of ADPKD families show *PKD1* mutations, while 15–20% show *PKD2* mutations [5]. Patients with *PKD2*-related ADPKD typically present with milder disease and experience a later onset of ESKD compared to those with *PKD1* mutations. As a result, younger patients and those with less aggressive presentations of *PKD2*-related ADPKD might go undiagnosed [4].

### 1.1. ADPKD Pathogenesis

The pathogenesis of ADPKD involves several mechanisms [6]. PC1 is a transmembrane glycoprotein receptor found in different cell types. In epithelial tubular cells, PC1 is located in the primary cilia. These organelles protrude from the cell surface and function as sensors that transmit extracellular signals into the cell (Figure 1) [5]. PC1 works as a mechanosensory that initiates cystogenesis and eventually senses the mechanical stimuli provided by the stiffness of the extracellular environment [7,8]. PC1 forms a complex containing E-cadherin and alpha-, beta-, and gamma-catenin. E-cadherin is a signaling center for cellular pathways and is constantly trafficked through an endocytic recycling pathway, which regulates cell adhesion and polarity [7,9]. PC2 is a multi-pass membrane protein that functions as a non-selective cationic channel permeable to calcium. It is involved in intracellular calcium homeostasis, channel activity of the endoplasmic reticulum, and mechanosensitive channel activity in the primary cilium of tubular epithelial cells [10]. PC1 and PC2 form a complex localized in the primary cilium of renal epithelial tubular cells and interact with each other to produce cation-permeable currents [11]. Other genes encoding components of the protein translocation apparatus of the endoplasmic reticulum membrane, such as *SEC63*, *SEC61B*, *GANAB*, *PRKCSH*, *DNAJB11*, *ALG8,* and *ALG9*, can diminish the functional dosage of PC1 by impairing its post-translational modification and trafficking to the cell membrane, modulating cystic disease severity in the setting of ADPKD [12].

In ADPKD, the hyperphosphorylation of PC1 causes PC1 and E-cadherin depletion from the cell membrane. This event alters PC1 localization, impairing its capacity to establish protein complexes that are crucial for the stabilization of adherent junctions and for the differentiation of the polarized renal epithelium [13]. Although demonstrated mainly in experimental models, polycystins alteration may contribute to ADPKD development through defects of ciliary function. The co-localization in primary cilia of polycystins with polaris and cystin, which are proteins that are disrupted in genetic mouse models of polycystic kidney disease (PKD), suggests that the alteration of polycystins may lead to abnormalities in the primary cilia, contributing to the pathogenesis of ADPKD [14]. Polycystins also regulate the structure and function of the primary cilia through the inhibition of the cystogenic gene Tubby-like protein 3 (Tulp3). Reduced expression of polycystins may promote cystogenesis through the activation of Tulp3 in late embryonic stages. In an adult murine model of PKD, the concomitant deletion of PC1 and Tulp3 ameliorated cystic disease. It is possible that Tulp3 controls distinct ciliary pathways that positively or negatively regulate cystogenesis depending on the cellular context [15]. These findings highlight the central role of ciliary and signaling dysfunction in the development and progression of ADPKD. However, the role of Tulp3 in the pathogenesis of ADPKD should be further investigated. Under physiological circumstances, polycystins are essential for the regulation of the differentiated phenotype of the tubular epithelium. Reducing polycystin expression below a critical threshold triggers a phenotypic transition marked by an inability to maintain planar polarity, heightened rates of proliferation and apoptosis, the induction of a secretory phenotype, and remodeling of the extracellular matrix [16].

In ADPKD patients, the activation of antidiuretic vasopressin V2 receptors stimulates the formation of cyclic adenosine monophosphate (cAMP), which is implicated in the growth of renal cysts by inducing tubular cell proliferation and secretion of fluid [17]. This fluid contains a lipid compound that stimulates both cAMP accumulation and cell proliferation while increasing transepithelial fluid secretion [18]. Furthermore, inflammation and oxidative stress also play a contributory role in the pathogenesis of ADPKD. The cysts themselves can trigger an inflammatory response and oxidative stress, and the accumulation of damaged proteins can promote cyst growth and kidney damage [19,20,21].

Finally, metabolic alterations can be seen in ADPKD. These include aerobic glycolysis, increased pentose phosphate pathway, glutamine anaplerosis, fatty acid biosynthesis, and decreased fatty acid oxidation and oxidative phosphorylation [22]. Intriguing studies demonstrated that mutation of *PKD1* results in enhanced aerobic glycolysis in the cells of polycystic kidneys. The metabolic inclination for aberrant cystic growth is oriented towards aerobic glycolysis, leading to elevated glutamine uptake and diminished oxidative phosphorylation. As a result, ADPKD cells shift their energy towards alternative pathways [23]. Glucose deprivation was associated with lower proliferation and higher apoptotic rates in *PKD1*-mutant cells compared to nondeprived cells [24].

### 1.2. ADPKD Clinical Presentation and Outcome

The first signs and symptoms of ADPKD typically manifest during adulthood, specifically in the third or fourth decade of life. However, presentation in childhood has also been reported. A “Two-Hit Model” has been proposed to explain the kidney phenotype observed in ADPKD patients. In fact, in the first two decades of the life of ADPKD patients, only a few cysts are usually detected in the kidneys, while hundreds of renal cysts may be found in most patients at the age of 50. There is also a large intra-familial variability in ADPKD [25]. According to the “Two-Hit Model”, if an individual has inherited a germline mutation (first hit), the development of additional cysts is not possible unless another mutation (somatic mutation) occurs in either the *PKD1* or *PKD2* loci (second hit) [26,27]. However, studies in murine models showed that *PKD1* inactivation in adult kidneys resulted in delayed cystogenesis [28,29]. These data suggest that a third hit is necessary to inactivate *PKD1*. Recently, it has been demonstrated that renal injury causes rapid cyst production in adult mice, supporting the hypothesis of a “Third-Hit Model” [30]. According to a “threshold model”, the cyst formation may be sustained by polycystin below a critical threshold because of *PKD1* and/or *PKD2* mutations, mutations of genes in the endoplasmic reticulum, or somatic mosaicism [12].

Clinically, the presentation of ADPKD may include back pain, gross hematuria, abdominal mass, urinary tract infection, and urolithiasis. Serum creatinine is normal until the age of 30–40, but an increase over the normal is commonly followed by a linear and irreversible progression to ESKD. Although ADPKD is marked by a progressive increase of bilateral renal cysts, eventually leading to ESKD, extra-renal manifestations are frequent. Today, ADPKD is considered a systemic disease (Figure 2). Cysts can develop in various epithelial organs, with the liver being the most common site (affecting 75% of patients) and less frequently in the pancreas, ovaries, choroid plexus, bronchi, and testicles. Liver cysts originate within the bile ducts and can give rise to infections or trigger hemorrhaging. In some cases, the proliferation of liver cysts can become extensive enough to exert a mass effect, necessitating surgical intervention. Arterial hypertension is a frequent complication that occurs in most patients. The pathogenesis of hypertension in ADPKD is complex. A decreased expression of *PKD1*/*PKD2* may lead to decreased nitric oxide levels, altered endothelial response to shear stress, and diminished vascular relaxation. The abnormal endothelial and vascular function, together with dysregulation of primary cilia and cyst expansion, results in intra-renal ischemia that activates the renin–angiotensin-aldosterone system (RAAS). The increasing number and size of cysts eventually lead to a further impairment of blood pressure and kidney failure [31]. A life-threatening complication is represented by the rupture of a cerebral aneurysm. The presence of unruptured aneurysms in ADPKD patients is not uncommon. In a large series of ADPKD patients who underwent magnetic resonance angiography screening, intracranial aneurysms were detected in 9% of patients [32]. Despite ongoing research, uncertainty remains regarding the appropriate management of asymptomatic aneurysms, although certain experts advocate for proactive treatment strategies [33]. Cardiac valvular abnormalities may also occur and include mitral valve prolapse, mitral regurgitation, aortic insufficiency, and tricuspid regurgitation. Left ventricular hypertrophy is common and has been observed even in normotensive individuals. Less frequent non-renal manifestations of ADPKD encompass colonic diverticula, hernias, and male infertility [34].

The long-term prognosis of ADPKD is usually severe since most patients eventually develop ESKD or die because of extra-renal complications. However, the progression of ADPKD is highly variable. Apart from exceptional cases of ADPKD in infancy, most patients enter a regular renal replacement treatment within the age of 60 years, but a minority of patients may experience kidney failure in their advanced elderly years. The severity of renal and extra-renal disease, the age of development of ESKD, and the risk of cerebral aneurysms can be different even among the members of the same family. It is not unusual for a son to enter ESKD earlier than the affected parent. Total kidney volume (TKV) measured by magnetic resonance imaging (MRI) represents a valuable prognostic indicator and enables the categorization of patients into those with slow or rapid disease progression, with significant implications for their treatment and care [35].

### 1.3. ADPKD Treatment

At present, there is no cure for ADPKD, but some measures may control symptoms and slow the growth of cysts. A healthy lifestyle should include physical exercise and weight control to prevent overweight and obesity, which are strongly and independently associated with kidney growth [36]. Salt intake should be reduced both to prevent hypertension and to avoid an increase in plasma osmolarity, which activates vasopressor V2 receptors. To evaluate whether sodium restriction may slow the progression of ADPKD, a post hoc analysis of two HALT-PKD clinical trials of RAAS blockade in ADPKD was made. In a study focusing on early ADPKD, both averaged and time-varying urinary sodium excretions exhibited significant associations with kidney growth, but they were not associated with faster estimated glomerular filtration rate (eGFR) decline. In the study on late ADPKD, the averaged but not time-varying urinary sodium excretion was significantly associated with increased risk for the composite endpoint (50% reduction in eGFR, ESKD, or death) and a significantly faster eGFR decline. These data suggest that sodium restriction is beneficial in the management of ADPKD [37]. In addition, a generous amount of plain water is recommended, at least in patients with normal kidney function. Water can reduce the plasma osmolarity, inhibiting the production of vasopressin V2 receptors and attenuating the deleterious effects of cAMP on the kidney cysts [38]. Arterial hypertension is the most important modifiable factor in ADPKD. Control of hypertension is critical to prevent the progression of kidney failure and cardiovascular events. RAAS inhibitors are considered as the most effective antihypertensive drugs. The efficacy of RAAS inhibitors has been evaluated in the two HALT-PKD clinical trials of RAAS blockade in patients with ADPKD. In the randomized controlled trial involving early ADPKD, 558 hypertensive individuals aged 15–49 with ADPKD and an eGFR of at least 60 mL/min/1.73 m^2^, participants were randomly allocated to receive either a standard blood pressure target (120/70 to 130/80 mm Hg) or a lower blood pressure target (95/60 to 110/75 mm Hg), along with either an angiotensin-converting-enzyme inhibitor (lisinopril) combined with an angiotensin-receptor blocker (telmisartan) or lisinopril combined with a placebo. The use of RAAS inhibitors compared with standard blood-pressure control obtained a slower increase in total kidney volume, no change in the eGFR, a greater decline in the left ventricular mass index, and a greater reduction in urinary albumin excretion [39]. In the trial on late ADPKD, 486 patients aged 18 to 64 with ADPKD and eGFR of 25–60 mL/min/1.73 m^2^ were enrolled. Participants were assigned to receive lisinopril or lisinopril and telmisartan at doses able to maintain blood pressure between 110/70 to 130/80 mm Hg. Monotherapy with lisinopril obtained blood-pressure control in most patients with ADPKD and kidney dysfunction. The addition of telmisartan did not modify the decline in eGFR [40]. In summary, these trials were unable to show the benefit of RAAS inhibitors on the deterioration of kidney function. However, these agents may reduce the burden of cardiovascular disease in patients with ADPKD. Moreover, one cannot exclude that a very early administration of RAAS inhibitors (or other antihypertensive agents) in patients with ADPKD and normal kidney function can slow the progression of the disease. Hyperuricemia is another modifiable risk factor. It is also deeply involved in hypertension and kidney failure [41]. Nevertheless, a post hoc analysis of HALT PKD concluded that elevated serum uric acid is not an independent risk factor for disease progression in ADPKD [42]. Once again, it is possible that trials enrolling younger patients with less advanced renal involvement could yield different outcomes compared to trials involving middle-aged individuals with longstanding or severe chronic kidney disease. Cyst infection is a frequent and serious complication of ADPKD. It is important to identify causative microorganisms; thus, blood and urine cultures should be performed before initiating antimicrobial therapy. Lipid-soluble antibiotics with good penetration into cysts are recommended. They include fluoroquinolones and trimethoprim–sulfamethoxazole. Serum creatinine should be checked during administration of these potentially nephrotoxic drugs in resistant cases; the drainage of large, infected cysts remains the main treatment [43].

Apart from symptomatic therapy, tolvaptan, a vasopressin V2 receptor antagonist, represents the only medication that has been demonstrated to attenuate ADPKD progression. In a multicenter, double-blind, placebo-controlled trial (TEMPO 3.4), 1445 patients ranging from 18 to 50 years of age, all diagnosed with ADPKD, possessing a total kidney volume exceeding 750 mL, and an eGFR surpassing 60 mL/min/1.73 m^2^, were randomly allocated to either receive tolvaptan at the maximum tolerable dose or a placebo. Over a 3-year follow-up, the increase in TKV in the tolvaptan group was 2.8% per year versus 5.5% per year in the placebo group (*p* < 0.001). The composite endpoint (worsening kidney function, kidney pain, hypertension, and albuminuria) favored tolvaptan over placebo (44 vs. 50 events per 100 years, *p* = 0.01). Tolvaptan was associated with lower rates of worsening kidney function (2 vs. 5 events per 100 person-years of follow-up). There were more discontinuations in the tolvaptan group, mainly related to aquaresis and hepatic adverse events [44]. To extend the study period by an additional 2 years and gather further data on the long-term safety and effectiveness of tolvaptan, 871 participants from TEMPO 3:4 were enrolled in TEMPO 4.4 trial. Percent changes in TKV after 2 years were not different (9.9% and 31.6%). Analysis of eGFR endpoints demonstrated a persistent effect on eGFR (3.15 mL/min/1.73 m^2^, *p* < 0.001). The safety profile of patients on tolvaptan in TEMPO 4:4 was similar to the one of TEMPO 3:4. It was concluded that the results of TEMPO 4:4 supported a modifying effect of tolvaptan on eGFR [45]. Another study (REPRISE trial) was done in patients with later stages of ADPKD. Participants were either 18 to 55 years of age with an eGFR of 25 to 65 mL/min per 1.73 m^2^ or 56 to 65 years of age with an eGFR of 25 to 44 mL/min per 1.73 m^2^. Patients were randomly assigned to receive tolvaptan or placebo for 12 months. The change from baseline in the eGFR was −2.34 mL/min in the tolvaptan group versus −3.61 mL/min in the placebo group (*p* < 0.001). Elevations in liver enzymes occurred in 38 of 681 patients in the tolvaptan group and in 8 of 685 in the placebo group. Enzymes returned to normal levels after stopping tolvaptan [46]. After the approval of Regulatory Authorities for the use of tolvaptan to slow the progression of cyst development and renal insufficiency in ADPKD, the algorithm to assess indications for initiation of treatment in ADPKD has been provided [47]. In summary, three multicenter, multinational trials showed that tolvaptan did not obtain a substantial effect on TKV but demonstrated a significant benefit in slowing kidney function deterioration both in early and late ADPKD. This benefit was maintained for at least 4 years. However, the aquaretic effect of tolvaptan is associated with polyuria, nocturia, polydipsia, and thirst, which are poorly tolerated by some individuals. Idiosyncratic hepatotoxicity can also occur, although it may be reversible on discontinuation of the drug. Although tolvaptan is the first drug showing significant benefit in the treatment of ADPKD, it requires careful attention in assessing the pros and cons of its use in ADPKD.

Somatostatin is a peptide secreted by pancreatic cells that may interfere with cyst development by different mechanisms, including the inhibition of cAMP generation [48]. Octreotide, a long-acting synthetic analog of somatostatin, was tested in a multicenter trial in adults with ADPKD and eGFR of at least 40 mL/min per 1·73 m^2^. Participants were randomly assigned to a 3-year treatment with octreotide or 0·9% sodium chloride solution every 28 days. Among them, 38 patients in the octreotide group and 37 in the placebo group had evaluable MRI scans at 1 year. TKV increased less in the octreotide group than in the placebo group (46.2 vs. 143·7 mL, *p* = 0·032). At 3 years, 35 patients in each group had evaluable MRI. The mean TKV increase in the octreotide group (220.1 mL) was smaller than in the placebo group (454.3 mL), although this difference did not reach statistical significance. The distribution of participants experiencing serious adverse events was similar in both treatment groups. However, the octreotide group had four cases of cholelithiasis or acute cholecystitis [49]. In another study on individuals with ADPKD and an eGFR between 15 and 40 mL/min, 100 participants were randomized to similar treatment with octreotide or placebo. At 3 years, TKV was reduced by 422 mL compared to placebo (*p* = 0.002). Conversely, the median rate of GFR decline did not exhibit statistical significance. Over a median 36 months of follow-up, 9 patients on octreotide and 21 on placebo progressed to a doubling of serum creatinine or ESKD [50]. A recent small, randomized trial in 19 participants with ADPKD showed that 4 weeks of treatment with octreotide combined with tolvaptan may reduce eGFR more effectively than octreotide and placebo. Octreotide reduced total and cystic kidney volumes and attenuated the aquaretic effect of tolvaptan [51]. This trial was too small, and the outcome was too short to allow conclusions. It is possible to consider whether the diminished aquaretic impact of tolvaptan, when combined with octreotide, might compromise the long-term efficacy of tolvaptan.

The administration of 2-Deoxyglucose (2DG) inhibited aerobic glycolysis in PKD mouse models and resulted in lower kidney weight and volume, cystic index, and proliferation rates compared to untreated mice [24]. Further studies outlined the safety of 2DG and confirmed its efficacy in inhibiting aerobic glycolysis and in slowing the progression of kidney disease in preclinical models of ADPKD [52,53,54].

## 2. Autophagy and ADPKD

### 2.1. Autophagy

Autophagy is a cellular recycling system that allows the reuse of old and damaged cell parts. There are three types of autophagy: macroautophagy, microautophagy, and chaperon-mediated autophagy, but the term autophagy usually indicates macroautophagy. These categories of autophagy are interconnected, albeit employing distinct protein types and diverse membrane dynamics.

Macroautophagy is a complex process for the lysosomal-dependent degradation of unwanted cellular material. In response to endogenous or exogenous stimuli, the autophagic activator complex forms a double membrane complex that sequesters the cargo (phagophore) and elongates, producing spherical vesicles (autophagosomes). These vesicles reach and merge with lysosomes, where the cytoplasmic material undergoes degradation (autolysosomes). Subsequently, the resultant macromolecules are released and recycled within the cytosol for further use [55]. Autophagy is executed by autophagy-related (Atg) genes. At present, 36 Atg proteins involved in autophagy have been identified. Two kinases regulate autophagy: the adenosine monophosphate-activated protein kinase (AMPK) and the mammalian target of rapamycin (mTOR) (Figure 3). AMPK is an enzyme that promotes glucose and fatty acid uptake and enhances their oxidation when cellular energy levels are depleted. AMPK stimulates autophagy by phosphorylating ULK1 (Unc-51-like kinase 1), which is part of a protein complex containing Atg13, Atg101, and FIP200 that initiates autophagy. The ULK1 phosphorylates and activates the protein Beclin-1, which is part of another complex that contains the proteins PIK3R4, Atg14L, and phosphatidylinositol 3-phosphate kinase. The ULK and Beclin-1 complexes contribute to the activation and regulation of autophagy. The mTOR is the subunit of two molecular complexes: mTOR complex 1 (mTORc1), or raptor complex, and mTOR complex 2 (mTORc2), or rictor complex. The mTORc1 is composed of a serin-threonine kinase, the regulator protein G (Rheb), and other proteins. The mTORC1 complex regulates cell growth (size), proliferation, translation, and autophagy, while mTORC2 controls the actin cytoskeleton and apoptosis [56]. Contrary to AMPK, mTOR inhibits autophagy by preventing the activation of ULK1 [57,58]. ULK1, in conjunction with the negative feedback loop involving mTOR and AMPK, plays a pivotal role in ensuring the precise functionality of cellular response mechanisms. AMPK downregulates mTOR, whereas mTOR inhibits AMPK; this inhibition is essential to maintain AMPK in an inactive state under normal physiological conditions [59].

Microautophagy is a nonselective form of lysosomal degradation usually induced by nitrogen starvation or by an AMP-activated protein kinase (AMPK)-regulated endosomal sorting complex required for transport [60]. The useless cytosolic substrate (cargo) is directly engulfed by lysosomes through protrusions or invaginations of the cargo membranes [61]. Microautophagy can collaborate with macroautophagy in organelle homeostasis by eliminating misfolded proteins and regulating the size of the endoplasmic reticulum, which synthesizes and exports proteins and membrane lipids, with mechanisms depending on cell type and cell function [62].

Chaperon-mediated autophagy is a selective form of autophagy that degrades cytosolic proteins. The signaling mechanisms governing chaperon-mediated autophagy are not yet fully elucidated, although they may encompass responses to stress conditions, including oxidative stress. In chaperon-mediated autophagy, irregularly unfolded proteins bearing a particular motif are recognized by the heat shock cognate 71 kDa protein, complexed with co-chaperones, and finally internalized in lysosomes via the lysosome-associated membrane protein 2 (LAMP2) receptor [63]. Recent studies demonstrated that chaperon-mediated autophagy may be involved in the execution of ferroptosis, a form of programmed cell death dependent on iron and characterized by the accumulation of lipid peroxides [64].

### 2.2. Autophagy in PKD Models

Several studies have highlighted the dysregulation of autophagy in PKD, demonstrating both increased and decreased autophagy activity across various experimental models. Impaired autophagy can contribute to the accumulation of damaged cellular components and cyst formation, while augmented autophagy may worsen cyst growth.

Polycystins PC1 and PC2 play crucial roles in ensuring proper autophagic processes. PC1 regulates calcium-dependent calpain proteases and maintains lysosomal integrity [65], while PC2 forms a complex with beclin-1, a key protein involved in autophagic vacuole formation [66]. The interplay between autophagy and apoptosis has been observed in murine PKD models, where suppressed autophagic flux correlates with increased apoptosis, leading to tubular epithelium proliferation and subsequent cyst formation [67]. It has been shown that PC2 is a critical mediator of autophagy induction [66,68] and that basal autophagy is enhanced in PC1-deficient cells, implying that PC1 promotes autophagic cell survival [69]. Malfunction of PC1 is associated with renal cyst development and declined renal function. Experimental studies in animal models of PKD showed that steviol, a major metabolite of the sweetening compound stevioside, can slow cyst progression in renal epithelial cells by enhancing and stabilizing PC1 protein expression and by promoting lysosomal degradation of β-catenin [70].

There is a reciprocal interaction between autophagy and primary cilia. In a study, signaling from the cilia induced autophagy, while inhibition of autophagy increased cilia growth [71]. In another study, autophagy induced primary cilium biogenesis by degrading the ciliopathy protein OFD1 (OFD1 centriole and centriolar satellite protein) at centriolar satellites [72]. In human kidney proximal tubular cells, mTOR activation was enhanced in cilia-suppressed cells, and MG132, an inhibitor of the proteasomes, could largely reverse autophagy suppression. These findings underscore a reciprocal regulatory relationship between cilia and autophagy [73].

The mTOR signaling pathway, which affects both autophagy and cilium signaling, is critical in cystogenesis [74]. In PKD cells with short cilia, mTOR is activated, and autophagy is repressed. Studies have demonstrated that mTOR inhibitors can slow cyst growth, reduce kidney volume, and improve kidney function in rodent models of PKD [75,76,77,78,79]. Double treatment with the mTOR inhibitor rapamycin and the cAMP inhibitor 2-Chloro-N(6)-(3-iodobenzyl) adenosine-5’-N-methylcarboxamide (Cl-IB-MECA) may have synergistic effects on the inhibition of cell proliferation in ADPKD [80]. Nevertheless, the findings from these studies are not conclusive. It has also been noted that the suppression of autophagy could potentially be associated with the creation and maintenance of cysts [81,82]. These differences have been ascribed to a variety of factors, encompassing variations in the models employed, diverse autophagic status, varying stages of PKD, and disparities in the timing, duration, and dosages of the administered drugs [83]. Overall, the available studies showed that dysregulated autophagy plays a significant role in PKD progression. Autophagy likely plays a multifaceted role in ADPKD, as illustrated in Figure 4. Experimental research has explored potential therapeutic approaches for PKD, including mTOR inhibitors such as rapamycin, although conflicting findings have indicated that rapamycin may exacerbate cyst growth. Other autophagy activators tested in murine PKD models include triptolide [84], statins [85], caspase inhibitors [86], cyclin-dependent kinase inhibitors [87], and steviol [88].

The available data suggest that activating autophagy may have a beneficial effect on reducing cyst growth in certain settings. However, it is important to note that in other experimental disease models, autophagy can promote cell death. Moreover, increased autophagic activity, even with its pro-survival functions, may have deleterious consequences. Excessive autophagy in fibroblasts has been shown to release collagen and contribute to tissue fibrosis, a common morphological feature observed in ADPKD [89]. The transforming growth factor beta 1 (TGF-β1) signaling pathway likely plays a significant role in this process by activating autophagy and inducing epithelial–mesenchymal transition in kidney cells. The intricate interplay between autophagy, fibrosis, and TGF-β1 signaling further highlights the complexity of autophagy’s involvement in ADPKD pathogenesis [90].

The exploration of autophagy regulators presents a significant challenge due to the intricacies involved in accurately measuring autophagy. A defining characteristic of autophagy is the formation of double-membrane autophagosomes, which can only be observed and characterized with electron microscopy. This technique allows researchers to visualize the distinct morphology of autophagosomes, providing direct evidence of autophagy initiation and progression [91]. However, relying solely on electron microscopy for autophagy measurement can be time-consuming and technically demanding [91]. To overcome these limitations, alternative approaches have been developed to complement or replace electron microscopy-based analysis. One such approach involves the utilization of molecular markers to track autophagy-related processes. A widely used marker is Green Fluorescent Protein-tagged light chain 3 (GFP-LC3), a protein that becomes associated with autophagosome membranes during their formation. By visualizing the fluorescence signal of GFP-LC3 using light microscopy, researchers can indirectly monitor autophagosome formation and estimate the extent of autophagy activity within cells. This technique allows for the examination of a large number of samples in a high-throughput manner, providing valuable insights into autophagy dynamics in various experimental settings [92]. Furthermore, recent advancements in high-throughput microscopy technology have revolutionized the field by enabling large-scale image-based screens using light microscopy. These cutting-edge techniques allow researchers to perform automated analyses of fluorescently tagged proteins or cellular structures associated with autophagy. By quantifying the changes in fluorescent signal intensity, distribution, or co-localization patterns, researchers can gain a deeper understanding of autophagy modulation and identify potential autophagy regulators or targets [93]. By integrating these complementary approaches, researchers can comprehensively assess autophagic flux, taking into account both autophagosome production and autophagosome-lysosome fusion.

### 2.3. Autophagy Regulators in ADPKD

Considering the ambivalent role of autophagy in ADPKD, therapeutic approaches involving autophagy inducers have shown conflicting results. Among the enhancers of autophagic flux are mTOR inhibitors and metformin. In a 2-year study, everolimus, compared to placebo, showed a slowing effect on the increase in total kidney volume in patients with ADPKD. However, it did not demonstrate a significant effect on the progression of kidney impairment [94]. A meta-analysis of five randomized trials in patients with ADPKD reported that there was no significant difference in glomerular filtration rate (GFR) between the mTOR inhibitor-treated and control groups, while proteinuria was significantly higher in the treated group than in the control group. The authors concluded that long-term treatment with mTOR inhibitors is not beneficial for patients with ADPKD [95]. The lack of benefit of mTOR inhibition may depend on the dosage. It is known that at standard doses, mTOR inhibitors can increase proteinuria and, in some cases, may also reduce GFR [96,97,98]. On the other hand, the complete inhibition of mTOR may lead to an overactivation of the autophagic flux, which can be harmful and potentially fatal to cells with preexisting proteostasis dysfunction [99]. It is possible that low-dose sirolimus or everolimus may prevent side effects and allow a limited activation of autophagy. Unfortunately, no trial on the effects of low-dose mTOR inhibitors on autophagy and on the course of ADPKD has been conducted.

Another drug that can enhance autophagic flux is metformin. This agent is mainly used to treat type 2 diabetes mellitus, but it can also activate AMPK, the master inducer of autophagy [100,101]. Experimental studies showed that metformin may prevent the progression of cysts and protect kidney function [102]. By increasing autophagy, metformin ameliorates the T helper 17 cells' inflammatory profile and improves mitochondrial bioenergetics [103]. In a 2-year randomized controlled trial involving 97 adults with ADPKD and an eGFR over 50 mL/min/1.73 m^2^, participants were assigned to receive either metformin or a placebo twice daily. The study observed an annual change in eGFR of −1.71 mL/min/1.73 m^2^ with metformin and −3.07 mL/min/1.73 m^2^ with the placebo. Side effects were comparable between the two groups. However, it is important to note that the reduction in eGFR decline with metformin was only modest and did not reach statistical significance [104]. In a 2-week randomized controlled trial, patients already taking tolvaptan were assigned to receive a placebo or hydrochlorothiazide and metformin. Combination treatment was superior to placebo on markers of disease progression (kidney weight, *p* = 0.003 and cystic index, *p* = 0.04) and reduced polyuria in comparison with tolvaptan [105]. The promising results of autophagy inducers observed in preclinical studies should be confirmed by robust randomized clinical trials to evaluate the long-term efficacy and safety of autophagy inducers in ADPKD. It is possible that autophagy has a multiple role in ADPKD. In fact, some experimental data may suggest that activating autophagy may reduce cyst growth. It should always be remembered that in certain experimental disease settings, autophagy may promote cell death. In other instances, even the pro-survival functions of increased autophagic activity may be deleterious. 

Vitamin D3 can activate autophagy and regulate cell proliferation by inhibiting oxidative stress and apoptosis [106,107]. In patients with ADPKD, low levels of serum 25-hydroxy vitamin D and vitamin D receptor expression are associated with a higher kidney volume [108]. These data may suggest a possible role of vitamin D3 in the management of ADPKD.

While waiting for the results of future studies, attempts to slow ADPKD progression may be made with combined treatments that associate the use of an autophagy regulator with drugs that can slow the ADPKD progression by other mechanisms, such as tolvaptan [44,46,109], octreotide [50], or 2DG. The association of metformin with 2DG could be particularly effective. A recent study demonstrated that the combination of metformin and 2DG blocked the formation of renal cysts and improved kidney function in ADPKD miniature pigs [110].

## 3. Conclusions

Autophagy plays a critical role in the pathogenesis of ADPKD; however, the impact of autophagy regulators in this context can be unpredictable, with potential outcomes that may be insignificant or even deleterious. As a result, combination therapy could be a path worth exploring for ADPKD treatment. Future trials focusing on the long-term safety and efficacy of combination treatments may incorporate an autophagy activator such as metformin or a low-dose mTOR inhibitor, along with a glycolysis inhibitor like 2DG, and/or an aquaretic drug such as tolvaptan. These comprehensive approaches have the potential to enhance therapeutic outcomes and address the complex mechanisms underlying ADPKD progression.

## Figures and Tables

**Figure 1 ijms-24-14666-f001:**
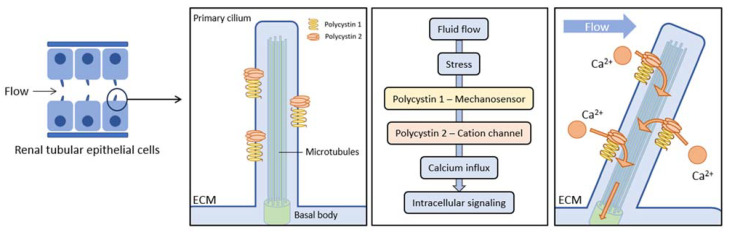
Primary cilia are small, hair-like structures that protrude from the surface of many cell types, including renal tubular cells in the kidneys. Primary cilia in renal tubular cells serve as sensory organelles that detect mechanical signals (fluid flow through the renal tubules). The bending of primary cilia in response to fluid flow activates the polycystin complex, which is composed of polycystin 1 (mechanosensor) and polycystin-2 (ion channel). This process induces the influx of calcium from extracellular matrix (ECM) into the cell, initiating cellular signaling processes.

**Figure 2 ijms-24-14666-f002:**
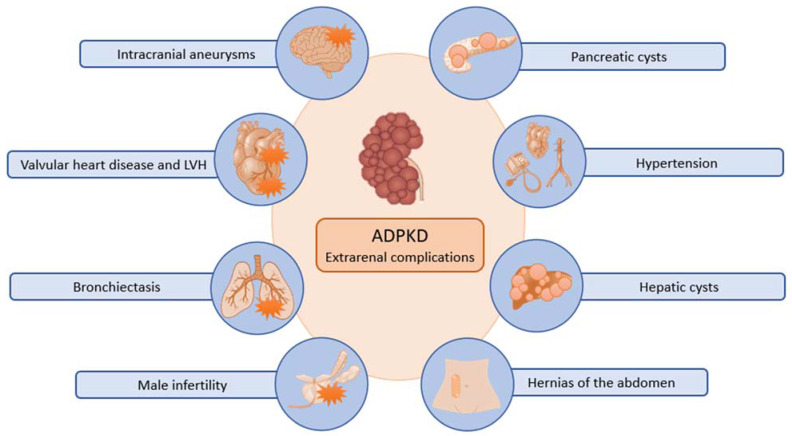
Major extrarenal complications of ADPKD.

**Figure 3 ijms-24-14666-f003:**
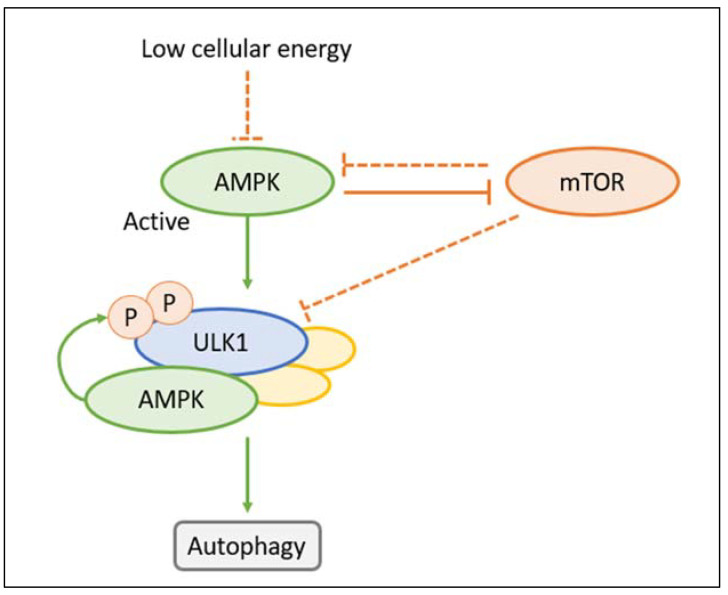
Low-cellular energy allows the activation of adenosine monophosphate-activated protein kinase (AMPK), which induces autophagy by phosphorylating Unc-51-like kinase 1 (ULK1). Contrary to AMPK, the mammalian target of rapamycin (mTOR) inhibits autophagy by preventing the activation of ULK1 in the case of normal cellular energy. There is negative feedback between mTOR and AMPK, which is critical to guarantee regulated autophagy. The mTOR inhibition of AMPK is required to keep AMPK inactive under physiological conditions.

**Figure 4 ijms-24-14666-f004:**
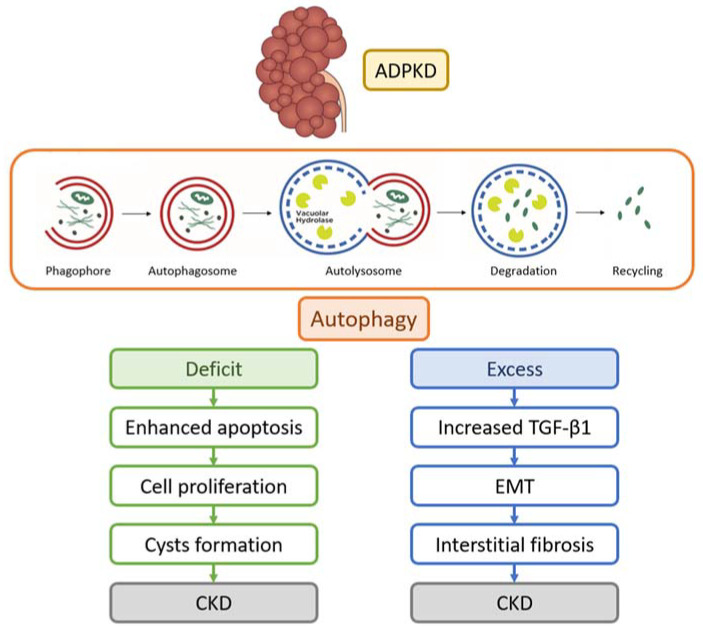
Macroautophagy is a highly regulated process of cellular degradation that removes and recycles unnecessary cellular material. The cytoplasmic material (cargo) is sequestered by a double-membrane structure, the so-called phagophore. Directed by autophagic genes, phagophore elongates around the cargo, producing a spherical vesicle with double-membrane termed autophagosome. The mature autophagosomes reach lysosomes and fuse with them (autolysosomes). Endocytic cargo and intracellular components sequestered by macroautophagy are finally degraded by the lysosome. The resulting macromolecules are released and recycled in the cytosol for reuse. In the case of defective autophagy, there is increased apoptotic activity, leading to kidney cell proliferation and cyst formation in Autosomal-Dominant Polycystic Kidney Disease (ADPKD), eventually leading to chronic kidney disease (CKD). In case of excessive autophagy, there is increased expression of transforming growth factor beta-1 (TGF-β1), which induces epithelial-mesenchymal transition (EMT), interstitial fibrosis, and CKD.

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
