# Peer review of "Autosomal Dominant Polycystic Kidney Disease: Is There a Role for Autophagy?"

_ijms, 2023, doi:10.3390/ijms241914666_

Round 1

Reviewer 1 Report

This is an extremely well written and insightful article on the possible role of Autophagy in ADPKD. This is of tremendous interest to biologists as well as clinicians who work on understanding the pathogenesis and natural history of this complicated condition.

Minor formatting issues where the font size is different- otherwise this reviewer is impressed by the manuscript.

none

Author Response

Dear reviewer, thank you very much for your kind comments. The formatting of the article has been revised to address the issues.

Thank you again

Reviewer 2 Report

Line 42: I don't agree with the statement that the mutation in PKD2 is more frequent in elderly compared to younger patients. I would rather write that age of ESKD caused by a mutation in PKD2 is higher compared to a mutation in PKD1.

Lines 115-122: I would expect that the Authors mention also "three-hit model" and "threshold model" of ADPKD pathogenesis [see:PMID: 27512790]

Line 141: "intracranial" would be better than "cerebral"

Author Response

Dear reviewer, thank you for the valuable advice.

Line 42: I don't agree with the statement that the mutation in PKD2 is more frequent in elderly compared to younger patients. I would rather write that age of ESKD caused by a mutation in PKD2 is higher compared to a mutation in PKD1. We have modified this sentence in accordance with your valid observation.

Lines 115-122: I would expect that the Authors mention also "three-hit model" and "threshold model" of ADPKD pathogenesis. A paragraph that mentions the three-hit model" and "threshold model" of ADPKD has been added. The reference has also been added.

Line 141: "intracranial" would be better than "cerebral" We have taken the advice into account, and the text has been modified accordingly.

Thank you again